# Performance of deep-learning-based approaches to improve polygenic scores

**Martin Kelemen** [1,2] ✉, **Yu Xu** [1,2], **Tao Jiang**[1,2], **Jing Hua Zhao**[1,2], **Carl A. Anderson**[3], **Chris Wallace** [4,5], **Adam Butterworth** [1,2,6,7,8] & **Michael Inouye** [1,2,6,7,9,10]

Polygenic scores, which estimate an individual's genetic propensity for a disease or trait, have the potential to become part of genomic healthcare. Neural-network based deep-learning has emerged as a method of intense interest to model complex, nonlinear phenomena, which may be adapted to exploit gene-gene and gene-environment interactions to potentially improve polygenic scores. We fit neural-network models to both simulated and 28 real traits in the UK Biobank. To infer the amount of nonlinearity present in a phenotype, we also present a framework using neural-networks, which controls for the potential confounding effect of linkage disequilibrium. Although we found evidence for small amounts of nonlinear effects, neural-network models were outperformed by linear regression models for both genetic-only and genetic +environmental input scenarios. In this work, we find that the usefulness of neural-networks for generating polygenic scores may currently be limited and confounded by joint tagging effects due to linkage disequilibrium.

Over the last two decades, genome-wide association studies (GWAS) have been deployed at scale to map genotype to phenotype. The success of these studies has driven a broad interest in using polygenic (risk) scores (PGS) to inform precision medicine[1]. PGS aggregates the signal from GWAS to produce a single value quantifying an individual's genetic predisposition for a disease[2] or trait. Powered by ever larger biobanks, PGS are anticipated to become part of genomic medicine across many health systems[3–5]. Indeed, PGS is already incorporated into some conventional clinical risk tools that have traditionally relied on non-genetic risk factors[6,7]. These non-genetic factors may contribute to disease risk directly and through interactions with genetic risk. These gene-environment interactions (GxE) arise when the genetic susceptibility to a disease or trait propensity is mediated by environmental variation, and a small number of robust GxE have been

identified[8–10]. Similarly, gene-gene interactions (GxG) arise through the combined effects of two or more alleles. The arising interdependence of the genome and the resulting nonlinear encoding of information has been acknowledged since at least Wright, who speculated on the importance of such effects on modelling a greater variety of phenotypic responses from a limited number of genes. Overall, despite their potential to improve disease risk prediction, higher-order nonlinear effects are seldom considered in current PGS models.

In parallel to the success of GWAS, neural networks (NNs) implemented through deep learning frameworks have achieved widespread adoption across many scientific disciplines. This surge in popularity is driven by algorithmic advances, significant increases in the volume of training data, and enhanced computational processing capabilities. Consequently, NNs have become the state-of-the-art approach for

[1]British Heart Foundation Cardiovascular Epidemiology Unit, Department of Public Health and Primary Care, University of Cambridge, Cambridge, UK. [2]Victor Phillip Dahdaleh Heart and Lung Research Institute, University of Cambridge, Cambridge, UK. [3]Wellcome Sanger Institute, Hinxton, Cambridgeshire, UK. [4]Cambridge Institute of Therapeutic Immunology & Infectious Disease, University of Cambridge, Cambridge, UK. [5]MRC Biostatistics Unit, University of Cambridge, Cambridge, UK. [6]British Heart Foundation Centre of Research Excellence, University of Cambridge, Cambridge, UK. [7]Health Data Research UK Cambridge, Wellcome Genome Campus and University of Cambridge, Cambridge, UK. [8]National Institute for Health and Care Research Blood and Transplant Research Unit in Donor Health and Behaviour, University of Cambridge, Cambridge, UK. [9]Cambridge Baker Systems Genomics Initiative, Department of Public Health and Primary Care, University of Cambridge, Cambridge, UK. [10]Cambridge Baker Systems Genomics Initiative, Baker Heart and Diabetes Institute, 75 Commercial Rd, Melbourne 3004 Victoria, Australia. ✉e-mail: mk907@medschl.cam.ac.uk

tasks in numerous fields, including the biomedical sciences[11]. The primary driver of NNs' success is their ability to model complex patterns, particularly where the outcome of interest depends heavily on higher-order nonlinear interactions that elude simpler methods[12].

Over the past decade, numerous studies have adapted NN approaches to address problems in genomics. In the field of sequence modelling, NNs have become the state-of-the-art tool for predicting the molecular consequences of changes in DNA or protein sequences[13]. Attempts to construct NNs to predict the effects of genetic variants on population-level trait variance have also been made[14–16]. Such efforts are attractive as GxG has been extensively demonstrated in model organisms and has been found in *cis* gene regulatory logic in humans[17]. Yet, while the PGS generated using NNs have generally shown substantive capacity for trait prediction, they often do not improve upon simpler additive models. Furthermore, there is conflicting literature around whether NNs have found genuine non-linear effects[14,18–23].

GxG or epistasis has two main forms, functional and statistical epistasis, which describe the two ways in which such interactions could contribute to the encoding of a phenotype[24]. Functional epistasis encompasses all inter-dependency of functionality between areas of the genome[25]. It is important to note that this only describes a joint mechanism that has been encoded in DNA, and this may not generate any observable phenotypic variance. Most NN models that demonstrated an improvement over other methods were primarily aimed at this task[26]. Statistical epistasis, on the other hand, describes deviations from additivity in a statistical model[27], which results in changes to phenotypic variation in a population. Statistical epistasis may also be caused by phenomena unrelated to biological mechanisms.

The correlation structure between different variants in the genome is known as linkage disequilibrium (LD)[28]. Such dependencies arise through the tendency of variants in close proximity to one another to be inherited together without recombination. The property of LD to 'tag' nearby causal variants has been exploited for GWAS[29], fine-mapping[30] and the development of PGS[31].

As the definitions of both LD and statistical epistasis require the co-occurrence of alleles[32], these two phenomena may perfectly overlap[33], which may then make differentiating these two challenging. For example, Hemani et al. reported many epistatic effects on gene expression[34] that were later revealed to be false positives caused by the two 'interacting' variants imperfectly and differentially tagging causal variants which had not been genotyped (Fig. 1). While this effect satisfies the mathematical definition of statistical epistasis, it is a technical artefact and thus unwanted as it does not indicate an encoding of additional information from deeper biological processes. For clarity, we refer to this type of statistical epistasis as *joint tagging effects* (also referred to as 'haplotype-effects' in Hemani et al.[35]).

To establish that genuine epistatic effects have contributed to phenotypic variance, it is necessary (but not sufficient) to show that fitted NN models improve over other additive models. Given prior issues regarding confounding by joint tagging effects, there is frequently a lack of evidence as to the nature of nonlinear effects, in particular that which differentiates between genuine epistasis and joint tagging effects.

Maximising the performance of risk prediction tools that harness all the available information across multi-modal genetic and environmental datasets is a key objective in order to realise the promise of genomic medicine. The rationales for previously observed performance gains by NNs and for continued widespread efforts to apply NNs to polygenic prediction are (i) the expectation that the extensive epistasis observed in model organisms also extends to humans, and (ii) statistical epistasis is the main driver of NN-based polygenic scores, which may then provide further scope of performance gains. In this work, we evaluate the latter rationale. We propose a framework to evaluate the nonlinear component of NN-based models, which we then apply in large-scale simulations and real data sets. Our study

contributes the following three key results. First, we present a way to address the joint tagging effect issue by weighting the SNP dosage input to NNs by LD-aware per-SNP PGS coefficients. Second, we demonstrate a way to find evidence for genuine epistatic effects by comparing NN models that have nonlinear capacity against those that do not. Finally, using these techniques on a large cohort of simulated and 28 real phenotypes, we quantify the contributions of GxG and GxE to phenotypic variance.

## Results
### Overview of the study
To understand how non-linear effects can contribute to phenotypic variation, which may be captured by NN-derived PGS, we performed a series of simulated and real data analyses. Phenotypes were assumed to arise from the model:

$$Y = G + E + G' \times E' + G' \times G' + E' \times E' + \epsilon \qquad (1)$$

where $Y$ represents the phenotype, and the additive terms of $E$, $G$ and $\epsilon$ denote the environment, genotype and noise components, respectively. The interaction terms $G' \times E'$, $G' \times G'$ and $E' \times E'$ denote the gene-environment, gene-gene and environment-environment interactions, where the $G'$ and $E'$ indicate subsets of genes and environmental predictors, respectively (Methods).

We used real genotype data from 125,000 European ancestry individuals from the UKB, partitioned in a ratio of 6:2:2 for training:validation:test sets (Methods). In the training and validation sets, we trained and selected NN models to generate PGS for both simulated and real phenotypes, with the top-performing model then evaluated on the held-out test set. To differentiate genuine epistasis from that generated by joint tagging effects, we applied a 'SNP-dosage weighting' strategy, which consisted of multiplying the LD-adjusted weights from the PGS Catalogue into the NN input. Finally, to assess the amount of non-linearity present, we compared the performance of NN models with and without activation functions enabled. All results are expressed relative to the LD-adjusted PGS, which means that negative values (for simulations) or values below 1.0 (for real data) indicate an unfavourable comparison between the NN models and their respective PGS baselines. For a graphical overview of the study, see Fig. 2.

### Simulation results
Simulation scenarios were generated on the same real genotype dataset as our real data analyses, using simulated phenotypes based on genetic architectures that consisted of entirely additive or entirely genuine epistatic effects. We chose to simulate these two extreme genetic architectures as, when compared, they allow the clearest evaluation of models and their ability to discriminate genuine epistasis from those that resulted from joint tagging effects masking a genetic architecture of a phenotype arising from purely additive effects. Importantly, it should be noted that higher-order interactions (e.g. 3-way and beyond) are expected to be statistically detectable in lower-order terms (e.g. 2-way interactions)[36], thus our analyses are expected to accommodate more complex (and mixed) epistatic scenarios. Additionally, we also simulated two mixed additive-epistatic scenarios. Here, the traits were generated by blending the phenotypes in the ratios of 2:1 and 1:2 from the additive:epistatic scenarios, respectively.

First, we demonstrated expected performances when constructing PGS from NNs in simulation scenarios where phenotypes were generated from an entirely additive architecture, but where we introduced joint tagging effects. Here, NN models capable of modelling nonlinearity ('nonlinear NN models') outperformed NN models that did not allow nonlinearity ('linear NN models') with a mean percentage change in $r^2$ of +1.6% ($P$-value = $2.7 \times 10^{-4}$). In this case, nonlinearities inferred by the nonlinear NN models were, in actuality, joint tagging effects, i.e. tagging additive effects which were being

missed by the linear NN models. Also, as expected, nonlinear NN models had better performance than linear NN models in scenarios that involved genuine epistatic interactions (mean % change in $r^2$ = +10.9%, *P*-value = 7.3 × 10⁻⁵), demonstrating the ability of non-linear NN models to leverage SNP-SNP interactions (Table 1), where they exist. The results for the additive-epistatic mixed scenarios were, as expected, intermediate between the purely additive and purely epistatic scenarios at 2.8% and 9.4% for the 2:1 and 1:2 additive:epistatic scenarios, respectively.

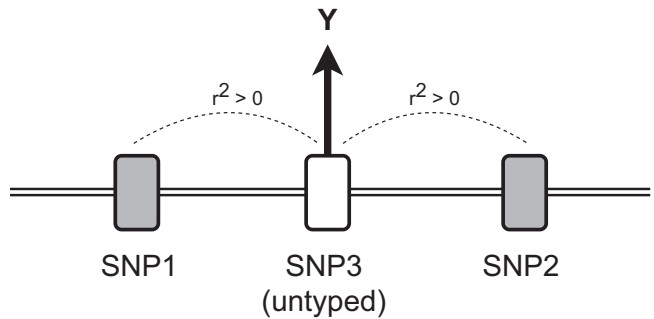

**Fig. 1 | Illustration of the joint tagging effect as an artefact generator for statistical epistasis.** Here, SNP1 and SNP2 imperfectly tag ($r^2$ > 0) the untyped SNP3, which is the causal variant affecting the phenotype Y. This pattern can arise even if the $r^2$ between SNP1 and SNP2 is zero. Statistical epistasis may only be generated by this pattern if the SNP1–SNP2 haplotype is a better tag for SNP3 than either SNP individually.

Surprisingly, as compared to linear NN models, nonlinear NN models had better performance in simulations comprising entirely additive effects (mean % change in $r^2$ = +3.7%, *P*-value = 4.4 × 10⁻¹⁰), i.e. no joint tagging effects were introduced. The fact that nonlinear NNs outperform linear NNs, even with no genuine epistasis or no joint-tagging effects present, has important implications for interpreting the results of the other scenarios. In the two scenarios where the phenotype signal arose entirely from genuine epistasis, with and without our mitigation strategy applied (10.9% and 9.5%), only the latter, smaller value should be considered to be measuring genuine epistasis. The difference between these two results in turn should then be interpreted as the nonlinear NN's ability to exploit LD to better model additive effects in the absence of any epistasis.

To address the problem of confounded nonlinear effects, we evaluated two joint tagging effect mitigation strategies. The 'SNP-dosage weighting' method (Methods) reduced joint tagging effects (difference in $r^2$: +0.2%, *P*-value = 0.40). On the other hand, the 'LD clumping + distance filtering' method appeared to overcompensate and predictive performance declined (difference in $r^2$: −1.2%, *P*-value = 4.4 × 10⁻¹¹). Applying the SNP-dosage weighting strategy to the scenario with true epistasis reduced the improvement slightly (mean percentage change in $r^2$ = 9.5%, *P*-value = 1.7 × 10⁻⁶), implying that little of the improved performance for nonlinear NN models in the presence of epistasis is due to joint tagging effects. Applying the 'SNP-dosage weighting' strategy also had a positive absolute effect on the predictive performance of all models, more so for epistatic than entirely additive scenarios (95.9% vs 65.1% increase in $r^2$, respectively).

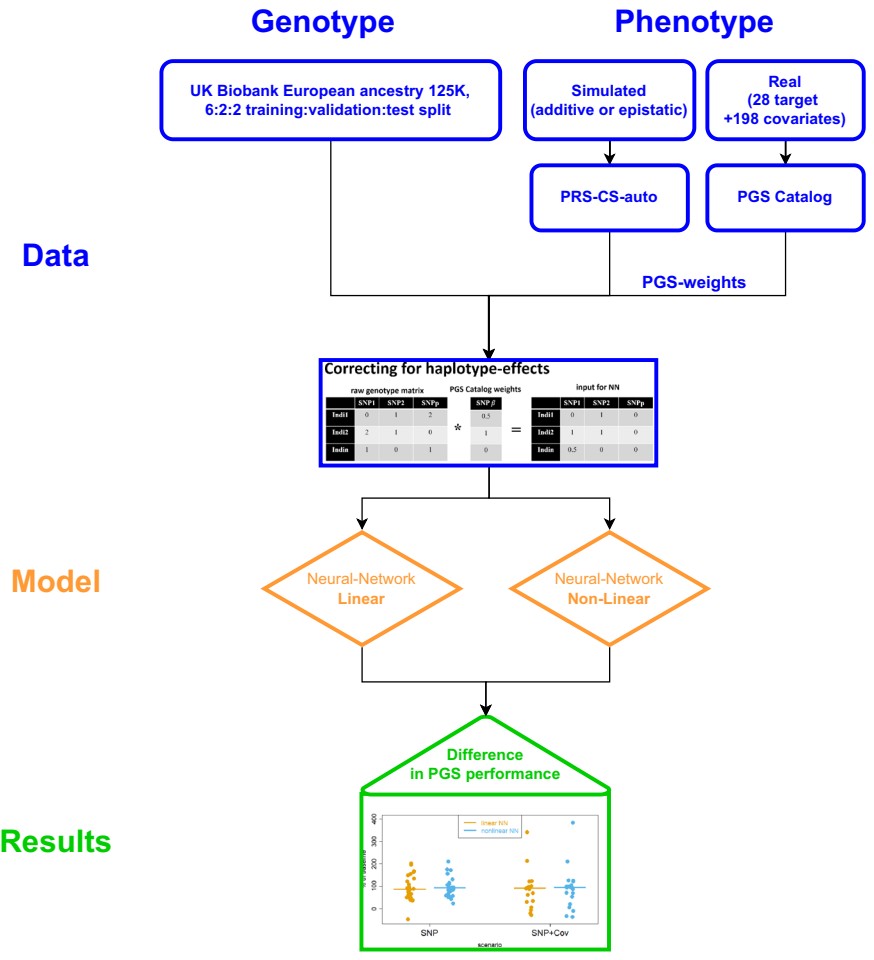

**Fig. 2 | Study schematic.** Flowchart that describes the data, models and results generation process, designated in blue, orange and green, respectively.

**Table 1 | Simulation results showing the difference between the nonlinear and linear NN model performance in the various scenarios**

| Epistasis | Additive | Joint tagging effects | Mitigation strategy | Nonlinear vs linear (%) | t-Test p |
|---|---|---|---|---|---|
| 100% | 0% | no | N/A | 10.9 | $7.271 \times 10^{-5}$ |
| 100% | 0% | no | SNP-dosage weighting | 9.5 | $1.749 \times 10^{-6}$ |
| 0% | 100% | no | N/A | 3.7 | $4.373 \times 10^{-10}$ |
| 0% | 100% | yes | none | 1.6 | $2.725 \times 10^{-4}$ |
| 0% | 100% | yes | SNP-dosage weighting | 0.2 | 0.398 |
| 0% | 100% | yes | LD clumping + distance filtering | −1.2 | $4.403 \times 10^{-11}$ |
| 33% | 66% | no | SNP-dosage weighting | 2.8 | $1.109 \times 10^{-4}$ |
| 66% | 33% | no | SNP-dosage weighting | 9.4 | $5.720 \times 10^{-5}$ |

Epistasis denotes the % of phenotypic variance due to genuine epistasis. Additive denotes the % of phenotypic variance due to additive effects. Joint tagging effects denote if the scenario involved artificially generating joint tagging effects (yes) or if there was complete coverage with no missing causal SNPs (no). The mitigation strategy shows the type of mitigation strategy deployed to address the joint tagging effects. Nonlinear vs linear shows the mean differences between the nonlinear and the linear NN models. To allow expressing the results of different phenotypes on the same scale, units were standardised in terms of % of the performance (evaluated by $r^2$) relative to an additive model from a PGS generated by PRS-CS. Positive values indicate that the nonlinear model performed better than the linear model, and conversely, a negative value indicates that the baseline additive PGS performed better. t-Test p shows the p-value of a (two-sided) paired t-test for the nonlinear vs linear mean difference. Source data is provided as a Source Data file.

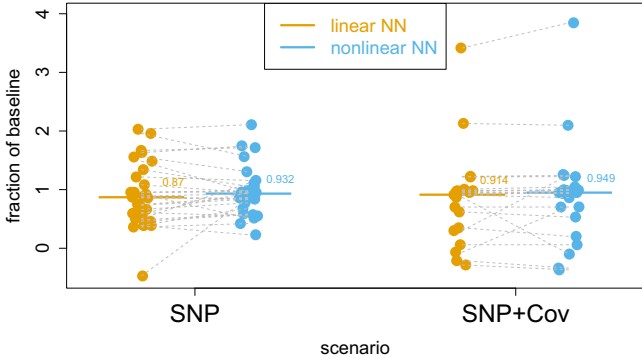

**Fig. 3 | Real data analysis results.** The bar shows the median, and each dot represents a result for neural-network model performance measured in $r^2$ between observed and predicted phenotype values, expressed as a fraction of their baseline linear regression models. Therefore, a value below 1.0 indicates that the NN model was outperformed by the LD-aware additive baseline. For the SNP scenarios, this was the PGS Catalogue PRS, and for the SNP + Cov scenarios, this was a multiple regression model that incorporated the PGS Catalogue PRS and the same covariates that were used in the neural-network models. Orange indicates linear and blue indicates nonlinear models. The grey dotted lines connect results that belong to the same phenotype. The medians were derived from 27 and 21 phenotypes for the SNP and SNP + Cov scenarios, respectively. Source data are provided as a Source Data file and also in Supplementary Tables 5 and 6.

## Application to the UK Biobank phenotypes

For the "SNP-dosage weighting" strategy, we utilised per-SNP PGS coefficients from the PGS Catalogue data, NN models were trained and evaluated on up to 28 phenotypes (Supplementary Table 3), utilising 125,000 European ancestry participants from the UK Biobank (Methods). NN models were trained with either of two input sets: SNP data only or SNP data and selected environmental variables (Supplementary Table 1). For each, NN models were fitted with and without activation functions enabled, i.e. with and without nonlinearities (Methods).

There were modest improvements in the performance of nonlinear NN models compared to linear NN models in the case of both SNP and SNP+environmental data inputs (6.86% and 3.77% median percentage change in $r^2$, respectively), suggesting the amount of nonlinearities attributable to GxG or GxE is limited (Fig. 3). However, the nonlinear NN models performed only 93.2% and 94.88% as well as their baseline linear regression models

(Methods) for the genetic-only and gene+environmental input scenarios, respectively.

As an illustrative example, consider the SNP input scenario for the cardio-metabolic traits. Here, the differences between the nonlinear vs linear NN models were 5.9%, −14.6%, 35%, 2.9% and 10.2% for ischaemic stroke, fasting glucose, venous thromboembolism, HbA1c and type 2 diabetes, respectively. So while on average, these nonlinear NN models had a 7.8% advantage over their linear counterparts, they still had a 37.2% lower mean performance than their additive PGS Catalogue baselines, indicating that the nonlinear effect is limited and may not be enough to offset the disadvantages of NN-based methods (Supplementary Table 5).

## Discussion

Deep learning based NN models have become the state-of-the-art tool in many areas, ranging from image classification[37], self-driving cars[38], playing video games[39] to modelling DNA sequence features[13]. The success of NNs in the aforementioned tasks has been attributed to their ability to exploit the nonlinear patterns inherent in those problems[12]. In this work, we examined the potential of NN-based approaches to exploit nonlinear effects due to GxG and GxE to improve the accuracy of polygenic prediction.

We found that joint tagging effects present substantial challenges in fitting NNs and interpreting their performance, as our nonlinear models outperformed linear models even in the absence of any genuine epistasis (Table 1). This result suggests that claims about the importance of epistasis based on NN performance need to be interpreted with caution without further evidence. It is also notable that even in the unrealistic scenario where a highly heritable trait was generated entirely from four-way interactions with no additive effects, the nonlinear NNs performed only ~10% better than the linear NN model. This result is expected from theory, which predicts that nonlinear trait variance is well captured by additive models, even in the presence of substantial epistatic effects[40]. Furthermore, we presented a SNP-dosage weighting strategy that can differentiate between joint tagging effects and genuine epistasis. This is useful, as the potential for NN-based improvements over and above LD modelling methods (e.g. LDpred) would be maximal if performance gains are due to genuine epistasis and not joint tagging effects. While for pure prediction tasks it is largely irrelevant whether a NN models joint tagging effects and not genuine epistasis, it is currently unlikely that a NN would outperform LD modelling methods at a given sample size. The competitiveness of NNs at modelling LD is also expected to decrease as the number

of SNPs is increased into the millions, as additive methods tend to scale better to higher-dimensional input data.

Notably, nonlinear NN models had a 3.7% advantage over linear NNs even in the absence of any deliberately generated joint tagging effects. The likely explanation for this is that through LD, the model was able to obtain better estimates of the true causal effects from nearby proxies than just from the main causal SNPs alone. This also suggests that among the two results that did involve genuine epistasis, the one with and one without 'SNP-dosage weighting' mitigation strategy applied to with results 10.9% and 9.5%, respectively, only the latter, smaller value should be interpreted to be due entirely to genuine epistasis.

We also sought to evaluate the potential of the NN-based approach in combined PGS+environmental factor models, which included up to 198 covariates (Supplementary Table 1). However, here we also found similar results, suggesting that gene-environment interactions contribute only low amounts (~4%) to trait variance. Explanations for this result could include that environmental risk factors in our study were either not evolutionary novel enough to generate interactions or the individual per-SNP-environment interactions were of too small effect to affect the overall prediction.

Our finding that nonlinear NNs outperform linear NNs, while simultaneously having lower performance than PGS-only models, suggests that while there may be some genuine epistatic contribution to phenotypic variation, the overall effect is limited and may not be enough to offset the disadvantages that NN-based methods face with the current generation of genetic data. NNs usually perform well in tasks where the signal-to-noise ratio is high, the input data conforms to regular spatial structures (to be exploited by convolutions), and where the sample size is not a limiting factor. In contrast, many of these requirements are not currently met for the task of phenotype prediction from SNP data (Supplementary Table 7). Furthermore, expectations of high NN performance for PGS generation rely on the assumption of the existence of a substantial (statistical) epistatic component of phenotypic variance. However, theory suggests that it is unlikely that large amounts of population-level phenotypic variance would be due to nonlinear effects[40], which would limit the potential of NNs for PRS generation even if more data became available.

Our study had several limitations, which may affect the interpretation of our results. Of the 28 examined traits that were available in the PGS Catalogue and met our QC criteria, 21 were cancer phenotypes, whose genetic architecture may differ from other phenotypes and diseases that arise from normal trait variation due to the importance of somatic mutations for cancers[41]. Alternative avenues by which NNs may contribute to phenotype prediction could be through combining multiple models in ensemble methods or through generating annotations (such as the presence of sequence features) that could be used for PGS generation. This latter route would allow NN-based methods to harness the abundant nonlinearity in functional epistasis in the genome sequence to generate annotation information that could be used to refine PGS, as has been demonstrated for autism spectrum disorders[42]. Our paper introduced two different mitigation strategies aimed at controlling the LD-driven 'joint tagging effects'. The practical benefits of the "SNP-dosage weighting" strategy will vary based on each individual dataset. On the other hand, the "LD-clumping + distance filtering" strategy is not practical in generating PGS, as it is a brute-force approach that eliminates most of the signal in its distance filtering step, therefore, it is only useful to show as a point of reference. Another limitation of our study was that all individuals in our analyses were of European ancestry, which was due to the poor representation of more diverse populations in the UKB. Finally, in the near future, the quantity of genomic data is expected to grow and reach the scale of tens of millions of individuals, including better representation of more diverse populations, also capturing their whole genome sequence at ever greater read depths[42,43]. Such an increase in the amount of data

may open avenues for newer NN-based approaches to search for statistical epistasis useful for trait prediction.

In conclusion, over the next decade, PGS is expected to be incorporated into genomic medicine via their addition into conventional risk prediction tools. While it is thought that NNs have the potential to substantially improve PGS performance, we found limited supporting evidence currently. Instead, we found that joint tagging effects and low levels of nonlinearity to be significant challenges for NN-based PGS generation. While NNs have become the state-of-the-art tool in many domains, expectations that NNs would also outperform additive models for PRS generation have not yet been realised.

## Methods

### Genotype processing steps

UK Biobank (UKB) is a large prospective cohort study with genetic and health-related data collected on ~500,000 individuals aged between 40 and 69 years old living across the United Kingdom. The initial data processing steps of this dataset are described in detail in its original publications[44,45].

Given our computational resources and the likely availability of data from the PGS Catalogue, the HRC + UK10K imputed genome dataset's variants were filtered to only keep HapMap3 SNPs, MAF > 0.001, maximum missingness per marker of 0.02, INFO > 0.9 that did not fail QC in any of the UKB batches. This process left 1,188,672 SNPs for further analyses. Samples were filtered to only keep unrelated (kinship coefficient <0.0884), European ancestry individuals with concordant recorded and inferred sex. To accommodate the scale of analyses on the available computing resources, this list of individuals was further filtered to 125,000 European ancestry individuals for each phenotype by keeping all cases (for binary traits) and prioritising to keep from the rest of the samples those with the most complete phenotype information. Datasets were then split in a pattern of 6:2:2 for training:validation:test sets.

### Covariate selection for the real data analyses

A total of 83 numeric or binary covariates and 115 multi-level factor covariates were chosen as predictors (Supplementary Table 1). These were then further subset into 10 phenotype categories, each defined by an exclusion criterion to remove covariates that could potentially overlap with the given target phenotypes. For example, for cancer-like target phenotypes, all cancer covariates were excluded, and for height, waist/height ratio and BMI were excluded. For a complete list of exclusions, see Supplementary Table 2.

### Selection of phenotypes and generating the PGS

Target phenotypes were selected by scraping the PGS Catalogue[46] database with the following criteria. Eligible PGS needed to have a training set of at least 50,000 individuals that excluded the UKB (to avoid data leakage) and were constructed using an LD-aware PGS method. We defined LD-aware as a method that incorporated LD reference panels to model signal overlaps between markers due to LD. The full list of these methods was: DBSLMM[47], metaGRS[48], lassosum[49], LDpred[31], LDpred2[50] and PRS-CS[51]. Matching phenotypes were identified in the UKB. For sex specific phenotypes (for example breast or prostate cancer), the individuals were filtered to keep only the relevant sex. With the exception of the trait 'cigarettes smoked per day', for continuous traits, individuals were removed who had an absolute trait value 4 SDs away from the mean. Next, using the weights from the PGS Catalogue, the baseline PGS was generated using PLINK2's '--score' function for all selected individuals. Finally, phenotypes whose PGS did not reach at least nominal significance ($p < 0.05$) on the UKB test set were removed. This process selected a total of 28 phenotypes for our analyses. For a complete list and processing criteria, see Supplementary Table 3.

## Fitting baseline models and covariate filtering

Two baseline models were evaluated to match the two NN model input scenarios. The first model was the performance of the PGS Catalogue PGS on its own on the UKB test set to serve as a baseline for the NN models that only had SNP and PGS as input data. The second model was a multiple linear or logistic regression model to serve as a baseline for the NN models that had the PGS + covariates as input data. This baseline had as input the PGS Catalogue, PGS, and up to 83 numeric or binary and 115 multi-level factor covariates. This model was fit twice, the first time with all predictors, and a second time with filtering to keep only those covariates that had a valid (non-NA), significant association ($p < 0.05$) with the phenotype. This same list of covariates was then taken forward to be used for the matching NN models. The training and test sets for both of these models were used with the same 75,000 and 25,000 individuals that were later used for the NN models.

## Controlling for joint tagging effects

To investigate the potential to mitigate the confounding caused by joint tagging effects, the following two strategies were evaluated:

1. SNP-dosage weighting: This entailed multiplying the allele dosages of the SNP input data fed to the NN by the PGS' per-SNP weights. We note that this strategy would not eliminate apparent statistical epistasis induced by joint tagging effects, instead, it is expected to reduce the performance gains possible due to such effects via the following mechanism. The extra information inferable from joint tagging effects that could be used to improve predictive performance would be present in the input, encoded in the additive SNP weights from LD-aware PGS generation methods. Therefore, when comparing the performance of nonlinear and linear NNs, where both models had these per-SNP PGS weights available, any advantage of the nonlinear NN over the linear NN would be likely due to genuine epistasis.

2. LD-clumping + distance filtering: This strategy entailed LD clumping performed in PLINK[52] on the GWAS results using '--tag-r2 0.5' and '--tag-kb 500' flags. Only a single variant from each locus was kept, and the surviving loci were then distance filtered, such that no two variants were allowed to be closer than 500 kb (always keeping the lower $p$-value signal). The rationale for this was that all surviving variants would be too far apart for LD to exist between them and untyped SNPs, and therefore for joint tagging effects to be generated. We note that this is a brute force approach, which may be only useful in eliminating the apparent statistical epistasis generated by joint tagging effects at the cost of eliminating the majority of the signal. Therefore, this strategy would be only viable to demonstrate the existence of genuine epistasis, but not for constructing state-of-the-art PGS models.

## Neural-network models

The NN models were custom-written in Python, utilising the PyTorch[53] framework. Multi-level factor covariates were transcoded into a one-hot representation, and continuous covariates (including individual-level PGS) were transformed into z-scores. All covariate input was then concatenated with the PLINK genotype data to be fed as input to the NN models. For simulations with the haplotype mitigation strategy 'SNP-dosage weighting', and for all real data analyses, the allele dosages of SNP input data [0,1,2] were multiplied by the per-SNP weights from the supplied PGS file. For case-control phenotypes, oversampling was enabled to achieve a 50:50 presentation of cases:controls to the model during training. The NN models were multi-layer perceptrons, with an architecture that consisted of three fully-connected hidden layers. Depending on the genetic architecture and the sample size available for a given phenotype, an overparameterized model with too many learnable weights may yield poorer performance.

Therefore, for each phenotype two models were fit, a 'large' model with 100, 50 and 25, and another 'small' model with 24, 12 and 6 neurons in each hidden layer, respectively. The rest of the model hyperparameters were batch size: 32, dropout: 0.3, learning rate: 0.001, batch normalisation enabled, softplus activation function and SGD optimiser. These model hyperparameters were based on a combination of literature review and preliminary tests using the random search via the hyperopt package[54,55], which included up to 20 hidden layers, 4000 neurons in the first layer, and a dropout of up to 0.9. For a more detailed explanation of model choice, see Supplementary Note 2. Models were trained until no performance improvement was observed on the validation sets for 12 epochs. Models that achieved their best validation performance in their first epoch were re-trained with the initial learning rate halved to further improve fit. For each phenotype, the final model was chosen to be either the 'large' or the 'small' one, based on their performance in the validation set as evaluated by $r^2$ between predicted and observed phenotypes.

## Evaluating the evidence for nonlinearity

The ability of a simple NN (without any special layer types, such as max pooling) to learn nonlinear phenomena is entirely due to the sequential application of a nonlinear activation function. Consider the following NN model containing $k$ hidden layers:

$$Y = \sigma_k(\ldots \sigma_2(\sigma_1(XW_1)W_2)\ldots W_k) \tag{2}$$

where $Y$ is a vector of continuous phenotypes, $X$ is the input data (such as SNPs), the $W_{1,2,k}$ are the model weights and the $\sigma_{1,2,k}$ are activation functions. This model can be transformed into a linear model with the same number of parameters by removing the activation functions as:

$$Y = \cancel{\sigma_k}(\ldots \cancel{\sigma_2}(\cancel{\sigma_1}(\mathbf{X}\mathbf{W_1})\mathbf{W_2})\ldots W_k) \tag{3}$$

This is true, as when the terms in the brackets are multiplied out, we recover the simple matrix multiplication of a linear model:

$$\begin{aligned} &= (((XW_1)W_2)\ldots W_k) \\ &= X(W_1W_2\ldots W_k) \\ &= XW_{\text{all}} \end{aligned} \tag{4}$$

However, switching off the activation function of an already trained model may not be optimal. Therefore, for our analyses, both linear and nonlinear models were trained from scratch according to the schema described above to ensure that optimal weights were learned in both cases. The difference in the performance between the nonlinear and linear models provides an estimate of the amount of nonlinearity learned by the nonlinear model.

## Quality control of real data results

NN results are expressed as a percentage of the $r^2$ of the relevant baseline method's performance on the test set. This relevant baseline was the PGS for the SNP-only, and the PGS + covariate multiple regression for the SNP+environment input scenario. To filter out low signal phenotypes that may generate outliers, results 500% away from the mean or where not all three models (linear, nonlinear or baseline) were at least nominally significant ($p < 0.05$) were removed.

## Simulation scenarios

To demonstrate the ability of NNs to leverage nonlinear effects to improve performance where genuine epistasis exists, and also to show the effects of the mitigation strategies that aimed to address the confounding issue caused by joint tagging effects, a series of simulation scenarios was performed. 125,000 European ancestry individuals and a random subset of 500,000 SNPs of the same data that was used

for the real data analyses were used to simulate 20 independent replicates of continuous traits. The genetic architectures of these traits were made up from 2000 causal variants in two scenarios: trait variance made up from entirely additive effects, and trait variance was made up entirely from 4-way epistatic interactions. In both cases, the total phenotypic variance was set to appear to have come from a narrow-sense $h^2$ of 0.5 (see Supplementary Note 1 for details). Although theory suggests that modelling a Dth order interaction is sufficient to capture signal from all nested lower order interactions[36]. We also examined mixed additive-epistatic scenarios. We generated additive-epistatic mixed scenarios by blending between additive and epistatic phenotypes in the ratios of 2:1 and 1:2 for additive:epistatic phenotypes, respectively.

Additionally, to simulate the confounding problem arising due to joint tagging effects, an additional sub-scenario was simulated starting from the phenotypes simulated from the trait architecture entirely due to additive effects. First, the squared correlation ($r^2$) was computed between all causal variants. Then, from those causal SNPs that had at least two other tagging variants with an $r^2 > 0.25$, up to 50% were randomly removed to simulate incomplete coverage of causal SNPs. This 'joint tagging effect reduced panel' recreated the situation where the true signal was missing, but was inferrable from two or more SNPs in LD with the causal variant.

To generate the baseline additive PGS, a GWAS was performed using PLINK2 on 60% of the individual genotype data, and PRS-CS auto was applied in two scenarios: once on the complete, 500,00 SNP panel, and another on the joint tagging effect reduced panel. After this process, using the entire surviving SNP panel, models with the same architecture as our real data analyses were trained independently.

### Reporting summary

Further information on research design is available in the Nature Portfolio Reporting Summary linked to this article.

## Data availability

This research has been conducted using the UK Biobank Resource under Application Number 7439. Data access policies (http://www.ukbiobank.ac.uk/register-apply/) and a description of the genetic data (http://www.ukbiobank.ac.uk/scientists-3/genetic-data/) are available from the UK Biobank website. The Research Analysis Platform is open to researchers who are listed as collaborators on UK Biobank-approved access applications. Data and scripts to reproduce figures and tables are provided in the Source data provided with this paper. Polygenic score data from the PGS Catalogue is publicly available from https://www.pgscatalog.org/. For the purpose of open access, the author has applied a Creative Commons Attribution (CC BY) licence to any Author Accepted Manuscript version arising from this submission. Source data are provided with this paper.

## Code availability

Software programs used in this study are all publicly available; PLINK2 and PLINK v1.9 can be downloaded from Christopher Chang's website [https://www.cog-genomics.org/plink/], PyTorch (v1.9.0+cu111) from the PyTorch website [https://pytorch.org/], R (v4.3.1) from the CRAN website [https://cran.r-project.org/] and Rstudio from the posit website [https://posit.co/products/open-source/rstudio/]. Code to perform all analyses reported in this manuscript is available at GitHub [github.com/mkelcb/dl-prs-paper] and Zenodo [https://doi.org/10.5281/zenodo.15324037][56].

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

## Acknowledgements

The views expressed are those of the author(s) and not necessarily those of the NHS, the NIHR or the Department of Health and Social Care. M.K. is funded by the BHF Cambridge CRE (RE/18/1/34212). This research was supported by the NIHR Cambridge Biomedical Research Centre (NIHR203312). The views expressed are those of the authors and not necessarily those of the NIHR or the Department of Health and Social Care. This research was funded by the MRC (MR/R013926, MC_UU_00002/4, MC_UU_00040/01), Wellcome Trust (WT220788) and supported by the NIHR Cambridge Biomedical Research Centre (NIHR203312), BHF Chair Award (CH/12/2/29428) and by Health Data Research UK, which is funded by the UK Medical Research Council, Engineering and Physical Sciences Research Council, Economic and Social Research Council, Department of Health and Social Care (England), Chief Scientist Office of the Scottish Government Health and Social Care Directorates, Health and Social Care Research and Development Division (Welsh Government), Public Health Agency (Northern Ireland), British Heart Foundation and the Wellcome Trust. M.I. is supported by the Munz Chair of Cardiovascular Prediction and Prevention and the NIHR Cambridge Biomedical Research Centre (NIHR203312) [*] as well as by the UK Economic and Social Research 878 Council (ES/T013192/1). This research was funded in part by the Wellcome Trust [Grant numbers 206194 and 108413/A/15/D].

## Author contributions

M.K. conceived and designed the study and performed all experiments. C.A.A. and C.W. contributed supervision and ideas for the analyses. M.K. and M.I. wrote the paper, with M.I. also contributing additional supervision for the analyses. Y.X., T.J., J.H.Z., and A.B. contributed feedback and suggestions to the paper.

## Competing interests

M.I. is a trustee of the Public Health Genomics (PHG) Foundation, a member of the Scientific Advisory Board of Open Targets, and has research collaborations with AstraZeneca, Nightingale Health and Pfizer which are unrelated to this study.CW receives funding from MSD and

GSK and is a part-time employee of GSK. These companies had no involvement in the work presented here. CAA has received consultancy or lecturing fees from Genomics plc, BridgeBio and GSK. The remaining authors declare no competing interests.
