## [Transparent Peer Review file · Nature Communications]

Performance of deep-learning based approaches to improve polygenic scores

Corresponding Author: Dr Martin Kelemen

Version 0:

Reviewer comments:

Reviewer #1

(Remarks to the Author)

This paper investigates the effectiveness of deep-learning neural network (NN) methods for improving polygenic scores (PGS) used in predicting genetic predisposition to diseases or traits. The study presents a framework to assess nonlinearity while attempting to control for confounding effects due to linkage disequilibrium (LD). Through simulations and real data from the UK Biobank, the authors find limited evidence for genuine nonlinear effects stemming from gene-gene (GxG) and gene-environment (GxE) interactions. The authors find that although neural network models can capture some nonlinear effects, they are generally outperformed by traditional linear regression models. The study highlights that joint tagging effects due to LD confound the detection of true nonlinear interactions, limiting the utility of NN-based PGS in current genomic prediction tasks.

The paper is methodologically detailed, the writing is clear and comprehensive, and the topic is timely and relevant. The work is quite original, and I see no flaws in data analysis. However the paper suffers from over-interpretation and thus, more work is needed to support the claims made by the authors.

Specific remarks:

1. Lack of References to Prior Work on Neural Networks for PRS: The authors do not sufficiently acknowledge or reference previous studies that apply neural networks to polygenic risk scores (PRS). Works, such as those found in PubMed (e.g., <https://pubmed.ncbi.nlm.nih.gov/33009504/>, <https://www.nature.com/articles/s41598-024-62513-1>), are missing, despite the field having a substantial body of research.
2. Absence of Discussion on Previous Nonlinearity Studies: The paper overlooks prior research investigating the contribution of nonlinearity to PRS. Studies like (<https://www.nature.com/articles/s41598-024-62945-9>) provide insights that could contextualize the current findings.
3. Possible Data Leakage in UK Biobank Phenotype Evaluation: The evaluation on UK Biobank phenotypes is problematic due to data leakage. Many phenotypes are correlated, and since the authors trained on both simulated and real data, the "held-out" test phenotypes may overlap with the training set, compromising the validity of the results.
4. Limited Benchmarking Against Established Methods: The authors infer the contribution of nonlinearity by comparing neural networks (NN) with and without activation functions and via linear models. However, the small observed contribution could be due to suboptimality of these. A comparison with established tools like PRSice and LDpred, known for effective polygenic score prediction, is necessary.
5. Narrow Definition of Nonlinearity: The conclusions about nonlinearity are limited by the narrow scope of machine learning models tested. Neural networks represent only one approach, and they may not perform optimally in this context. To make broader claims about nonlinearity, the authors should test alternative models, such as boosted trees and/or support vector machines with various kernels.
6. Simplistic Simulation Scenarios: The simulations consider only two extreme scenarios: entirely additive or entirely genuine epistatic effects. This binary approach is overly simplistic and does not reflect the complex and mixed genetic architectures observed in real-world phenotypes.

7. Insufficient Exploration of Joint Tagging Mitigation Strategies: The paper introduces strategies like SNP-dosage weighting to mitigate joint tagging effects but does not fully explore their efficacy. A more comprehensive evaluation of these strategies and their impact on results is needed.

(Remarks on code availability)

Choice of Python Library Restricts Code Accessibility: Although the provided code appears functional, the authors use the niche Python library "knet". This choice limits accessibility and reproducibility for readers unfamiliar with this framework. Using more widely adopted libraries like TensorFlow or PyTorch would improve usability and facilitate peer review.

Reviewer #2

(Remarks to the Author)

This is an interesting paper, investigating the use of NN based methods in the generation of polygenic scores, and polygenic risk models that include other risk factors, to predict diseases or phenotypes in UK biobank.

I think this paper is well written. It has a more methodological focus than some applied papers, however I found it very easy to read, and that the authors had done a very good job of making potentially complex genetic concepts understandable. I think the paper is important as it highlights that novel NN methods that have offered improvements in many areas of science, have some potential trade offs for genetic analyses and may not be currently suitable for PGS generation at scale, and the authors do a good and balanced job of describing this.

The authors rigorously generate PRS, and NN models, to predict 28 phenotypes in UK biobank. They compare NN models that potentially account for nonlinear interactions (including gene-gene interactions and gene environment interactions). They first perform an analysis in a simulated dataset that compares traits that are best explained entered by a simple additive model of heritability, traits that include joint tagging interactions, and traits whose variance is explained by true epistasis (phenotypes that are at the opposite end of the spectrum - explained entirely by complex non linear interactions). I enjoyed this approach of simulating a positive and negative control dataset to check that their approach could distinguish between these two artificially created extremes, before then proceeding to an analysis of real data from 28 UKBB phenotypes.

Their analysis when comparing within the two NN approaches highlighted that the joint tagging effect is a very important consideration when considering whether observed statistical epistasis relates to joint tagging effects or genuine epistatic interactions.

I found the noise introduced by the NN approach compared to baseline models interesting. They highlighted that the baseline linear PGS models were more predictive than the NN approaches they used. This is an important "negative" and is reported clearly, I think this is an important study and message for the field.

I think the work supports the claims the authors make, the methods are sound, with some potential limitations,

Comments/questions

General

The authors highlight that they had certain selection criteria for traits to be included, is it possible that the traits and the limited population selected do not cover the full breadth of variation in disease heritability - and some diseases may have more GG or GE interactions than others, where NN models may have more relevance in helping elucidate genetic architecture.

Can they make general inferences about using linear PGS models v NN models based on these results? Are there criteria that would need to be met for NN models to outperform simpler methods, or do their results from simulated data highlight that even in the most extreme settings of genetic non linear interactions, there is no prediction advantage to their NN approach?

How generalisable are their findings?, and how fragile are the findings to their choice of parameters, discovery data, covariates, phenotypes, genetic data, ancestry etc....I think they could comment on this a little more in the limitations, or show more data to back up the generalisability. this is important in particular because the strength of their negative findings depends a little on the generalisability of their findings.

They used standard criteria to QC data and exclude covariates before model generation, however is there a danger that this could remove the chance for the NN models to find occult interactions in the data, not apparent during the QC procedures? I think this is computationally efficient, but could it also reduce the chance of finding new non-intuitive interactions within the data?

Did you consider other non NN based approaches to take non linear effects into account, given the good performance of baseline PGS models, could you consider approaches that build on the baseline methods?

Comparison of linear and nonlinear NN models when prediction simulated polygenic scores with entirely additive effects. Can they just explain again what the main result is here: "This result in turn suggests that non linear NN models are better able to infer (false) interaction effects that capture phenotypic variance than they are at inferring (true) additive effects."

I think non linear NN models seemed to offer more improvement over linear models even when there were no introduced introduced epistasis or joint tagging effects. I am not sure what the meaning of this is, and wonder if the authors could rephrase or re explain?

Figure 3

They report the average change in polygenic score performance when using NN models compared to PGS baseline models, there is a spread, with some NN models (particularly two outliers in the models including covariates) seeming to perform considerably better than the PGS baseline, is this just noise or does it reflect the fact that in a select subset of diseases,

(Remarks on code availability)

Reviewer #4

(Remarks to the Author)

This paper presents a comparison of linear and non-linear neural network models for genomic prediction in humans. No large advantage if any is found on using non linearities, overall. I cannot see a clear target of the study. If the goal is to assess NN performance, at least a linear standard method should be added, such as ridge regression. If the goal is to compare linear vs say epistasis, linear model such as GBLUP can be generalized to include a AxA matrices.

Methods are not clearly described and, sometimes, conclusions are reached without enough evidence. I agree that linkage disequilibrium can be confounded with epistasis, but a comparison between linear and non-linear NN does not provide a clear conclusion between what is mor prevalent. In fact, both can be present simultaneously.

The SNP dosage weight is not justified. As I understood initially, it required running an algorithm twice, the first time to obtain the weights (I assume only with the training set). It seems these are weights for SNPs obtained from previous studies, but how are these wights obtained, are of on same dataset? same diseases. If so, you will likely be using test and training data even if unknowingly.

But even so, what are the properties of this approach? May be an ensemble approach would be more clearly justified...

Additional comments

- Reference 45 to PGS catalog cannot be accessed.
- It seems multi layer perceprons are the only strategies used, what about CNNs?
- Methods are not enough detailed , a github or similar with documented code would help.

(Remarks on code availability)

No code available.

Version 1:

Reviewer comments:

Reviewer #1

(Remarks to the Author)

The revised manuscript addresses most of the concerns raised in the initial review. The authors have clarified their methodological framework, provided additional simulations incorporating mixed additive and epistatic effects, and more explicitly contextualized their comparisons to established LD-aware PGS methods. The focus on deep learning methods is now better justified, though the exclusion of other nonlinear modeling approaches (e.g., tree-based ensembles or kernel methods) remains a limitation. While the authors argue this is beyond the scope of the present study, a broader comparative analysis would be necessary to draw general conclusions about the role of nonlinearity in polygenic prediction.

Nevertheless, the analyses presented are rigorous, the findings are clearly reported, and the manuscript represents a useful contribution to ongoing discussions about the utility of neural networks for PGS construction. I have no further major objections to publication.

(Remarks on code availability)

Didn't try to run it for obvios reasons, but looks fine.

Reviewer #2

(Remarks to the Author)

the reviewers have answered my questions appropriately and I have no further comments

(Remarks on code availability)

Reviewer #4

(Remarks to the Author)

I have no further comments, authors have replied to my concerns reasonably.

It is unlikely the code provided can be used by external users in a real data size problem, given its lack of documentation.

A requirement module list would be helpful to avoid installing new modules one by one when the original code is run and stops. The exact definition of `knet` would be `knet='scripts/python/Knet/Knet.py'` when running from within the 'dl-prs-paper-master' folder.

(Remarks on code availability)

Please find our itemised responses to the points raised by each reviewer below:

“Reviewer’s Comments:”

“Reviewer #1 (Remarks to the Author):

This paper investigates the effectiveness of deep-learning neural network (NN) methods for improving polygenic scores (PGS) used in predicting genetic predisposition to diseases or traits. The study presents a framework to assess nonlinearity while attempting to control for confounding effects due to linkage disequilibrium (LD). Through simulations and real data from the UK Biobank, the authors find limited evidence for genuine nonlinear effects stemming from gene-gene (GxG) and gene-environment (GxE) interactions. The authors find that although neural network models can capture some nonlinear effects, they are generally outperformed by traditional linear regression models. The study highlights that joint tagging effects due to LD confound the detection of true nonlinear interactions, limiting the utility of NN-based PGS in current genomic prediction tasks.

The paper is methodologically detailed, the writing is clear and comprehensive, and the topic is timely and relevant. The work is quite original, and I see no flaws in data analysis. However the paper suffers from over-interpretation and thus, more work is needed to support the claims made by the authors.”

We greatly appreciate the reviewer recognising the relevance of our work and also take on board that there were still a few areas where we needed to expand or explain our analyses to improve clarity.

“Specific remarks:

1. Lack of References to Prior Work on Neural Networks for PRS: The authors do not sufficiently acknowledge or reference previous studies that apply neural networks to polygenic risk scores (PRS). Works, such as those found in PubMed (e.g., <https://pubmed.ncbi.nlm.nih.gov/33009504/>, <https://www.nature.com/articles/s41598-024-62513-1>), are missing, despite the field having a substantial body of research.”

We thank the reviewer for bringing these papers to our attention, which we have now added as citations at the relevant part in the Introduction. AI is a broad, very fast moving field, so it is difficult to present a fully comprehensive review of it at any given time. Having looked at these publications now in detail, we can confirm that neither of these cover the main unique contribution of our own work, the acknowledgement and distinguishing between genuine epistasis and LD-driven joint tagging effects. Therefore, we believe that the main contributions of our paper remain unique.

"2. Absence of Discussion on Previous Nonlinearity Studies: The paper overlooks prior research investigating the contribution of nonlinearity to PRS. Studies like (<https://www.nature.com/articles/s41598-024-62945-9>) provide insights that could contextualize the current findings.

5. Narrow Definition of Nonlinearity: The conclusions about nonlinearity are limited by the narrow scope of machine learning models tested. Neural networks represent only one approach, and they may not perform optimally in this context. To make broader claims about nonlinearity, the authors should test alternative models, such as boosted trees and/or support vector machines with various kernels."

We recognise the value in contextualising our work and appreciate the suggested paper. However, we feel that broadening the subject of our paper to cover other methods and types of nonlinearities would be beyond the scope of our target in this work. The purpose of our paper is to evaluate the potentials of deep learning in exploiting nonlinearities for PGS generation, while accounting for the almost always neglected LD-driven confounding of joint-tagging effects. While the suggested methods are interesting and are tangentially related to the topics covered in our paper, we feel that a narrower focus would benefit readers looking for the topics we wanted to investigate here. Expanding investigations to other methods would be more appropriate to a broader methods benchmark / 'bake off' style work, certainly a worthwhile endeavour, but we believe that our focus on deep learning here is a strength and will be of high readership interest given its popularity.

"3. Possible Data Leakage in UK Biobank Phenotype Evaluation: The evaluation on UK Biobank phenotypes is problematic due to data leakage. Many phenotypes are correlated, and since the authors trained on both simulated and real data, the "held-out" test phenotypes may overlap with the training set, compromising the validity of the results."

While it is of course good practice to always be wary of potential overfitting, we can reassure the reviewer that we have taken a number of steps specifically aimed at eliminating the potential for any data leakage/overfitting:

- A. In our simulation analyses, all phenotypes were generated independently from each other and also from real data phenotypes. Therefore, they shared no genetic aetiology/correlation in their architectures that could have caused leakage.
- B. As for the real-data analyses, the PGS Catalog PGS summary data that were used as a basis for the NN training was specifically filtered to exclude any UKB samples. So, this step would have eliminated the potential for any leakage between our training and evaluation sets too.
- C. Finally, as for the situation where the real phenotypes themselves would be correlated due to a shared genetic architecture, we cannot see how this could have caused a problem of overfitting/leakage. For PGS model training, we split the data into training/validation/test sets by individual, not by genome. Therefore,

if the samples did not overlap, then such a leak could not ordinarily occur. The only situation we could imagine where this could have happened, if the different NN models shared weights, but this was not the case for us. This is because each NN model was trained independently, without sharing any model weights with other already trained models. Finally, if certain phenotypes were correlated with each other due to their shared genetic aetiologies, while this may have caused our results to be correlated, the model performance itself would have remained unbiased. It is also not standard practice in risk prediction to exclude individuals with similar phenotypes across training/validation/test sets; indeed, to do so risks creating unrealistic scenarios which would thus result in performance that would not be mirrored if the model was applied in the real world.

To make this clearer for the reader, we have re-edited the Methods to remove potential ambiguities at the following three areas:

*“Eligible PGS needed to have a training set of at least 50,000 individuals that excluded the UKB (**to avoid data leakage**) and were constructed using an LD-aware PGS method.”*

And:

*“125,000 European ancestry individuals and a random subset of 500,000 SNPs of the same data that was used for the real data analyses were used to simulate 20 **independent** replicates of continuous traits.”*

And:

*“After this process, using the entire surviving SNP panel, models with the same architecture as our real data analyses **were trained independently.**”*

“4. Limited Benchmarking Against Established Methods: The authors infer the contribution of nonlinearity by comparing neural networks (NN) with and without activation functions and via linear models. However, the small observed contribution could be due to suboptimality of these. A comparison with established tools like PRSice and LDpred, known for effective polygenic score prediction, is necessary.”

We thank the reviewer for bringing this to our attention. This issue is one of lack of clarity in explaining our methodology, rather than a weakness in our analyses. Our NN models, both linear and nonlinear, actually start from such relevant baselines the reviewer mentioned (e.g. LDpred). This is a crucial first step of our “SNP-dosage weighting” strategy, as we argue that by using such weights we can account for any ‘false’ epistasis due to joint tagging effects. However, what was not emphasised adequately in our initial draft was that as all NN results were expressed as a percentage of these baseline PGS methods, the comparison the reviewer has asked for is already part of our results. This meant, for example, if an NN result was less than 100%, then

that NN model performed worse than its baseline PGS, whereas if it was greater than 100%, then that meant that it outperformed the established baseline. This was stated under ***“Quality control of real data results”***, and the full list of baseline methods is mentioned under the Methods section, under the heading ***“Selection of phenotypes and generating the PGS”***:

“...were constructed using an LD-aware PGS method. We defined LD-aware as a method that incorporated LD reference panels to model signal overlaps between markers due to LD. The full list of these methods was: DBSLMM⁴⁶, metaGRS⁴⁷, lassosum⁴⁸, LDpred³⁰, LDpred2⁴⁹ and PRS-CS⁵⁰.”

However, we apologise that it was unclear to the reader that these comparisons were performed. Therefore, we have now explicitly spelled this out in our results section by adding the following text:

“All results are expressed relative to the LD-adjusted PGS, which means that negative values (for simulations) or values below 1.0 (for real data) indicate an unfavourable comparison between the NN models and their respective PGS baselines.”

The legend for Fig 3 now states:

“...expressed as a fraction of their baseline linear regression models. Therefore, a value below 1.0 indicates that the NN model was outperformed by the LD-aware additive baseline.”

And finally, the legend of Table 1 now also states:

“Positive values indicate that the nonlinear model performed better than the linear model and conversely, a negative value indicates that the baseline additive PGS performed better.”

“6. Simplistic Simulation Scenarios: The simulations consider only two extreme scenarios: entirely additive or entirely genuine epistatic effects. This binary approach is overly simplistic and does not reflect the complex and mixed genetic architectures observed in real-world phenotypes.”

The omission of an analysis investigating the mixed scenario of both additive and genuine epistatic effects was something we gave considerable thought to when preparing our study. At the time we decided that this analysis was not necessary, as theory indicates that our models detect higher-order interactions through modelling nested lower-order interactions (including main effects). That is, to detect an interaction between, say, SNPs 1, 2 and 3, the model also needs to detect all three two-way

interactions and all three main effects too. Therefore, the 'mixed scenario' that the reviewer requests is implicitly already covered by the entirely genuine epistatic effects scenario. This was demonstrated in Sorokina et al, 2008, and we further clarify this under both Simulation Results and Simulation scenarios:

"Importantly, it should be noted that higher order interactions (e.g. 3-way and beyond) are expected to be statistically detectable in lower order terms (e.g. 2-way interactions)³⁵".

In this revision we have now also added an empirical demonstration that this theoretical work holds true in practice. These simulation analyses to cover mixed scenarios have been added to the Results. In these new analyses we generated phenotypes that are a blend between additive and epistatic signals in the ratios of 2:1 and 1:2 for additive:epistatic phenotypes, respectively. With the addition of these scenarios, our results now have a gradient that incrementally goes from entirely additive to entirely epistatic phenotype at the increments of 0%, 33%, 66% and 100%. These new analyses further confirmed the theory: their results were intermediate between the two extremes. This is now added under the Results section:

"Additionally, we also simulated two mixed additive-epistatic scenarios. Here, the traits were generated by blending the phenotypes in the ratios of 2:1 and 1:2 from the additive:epistatic scenarios, respectively."

And:

"The results for the additive-epistatic mixed scenarios were, as expected, intermediate between the purely additive and purely epistatic scenarios at 2.8% and 9.4% for the 2:1 and 1:2 additive:epistatic scenarios, respectively"

And finally, also in the Methods section at:

"Although theory suggests that modelling a Dth order interaction is sufficient to capture signal from all nested lower order interactions³⁶, we also examined mixed additive-epistatic scenarios. We generated additive-epistatic mixed scenarios by blending between additive and epistatic phenotypes in the ratios of 2:1 and 1:2 for additive:epistatic phenotypes, respectively."

"7. Insufficient Exploration of Joint Tagging Mitigation Strategies: The paper introduces strategies like SNP-dosage weighting to mitigate joint tagging effects but does not fully explore their efficacy. A more comprehensive evaluation of these strategies and their impact on results is needed."

Our paper introduced two different mitigation strategies aimed at controlling the LD-driven 'joint tagging effects', described in detail in the Methods section, under "**Controlling for joint tagging effects**". The first of these is "**SNP-dosage weighting**" and the second is "**LD-clumping + distance filtering**". The second strategy is not practical in generating PGS, as it is a brute-force approach that throws away almost all of the signal (in the distance filtering step), so it is only useful to show as a point of reference.

The following two simulation scenarios are useful at examining if the "**SNP-dosage weighting**" strategy worked or not:

(1) Simulated phenotype involves no joint tagging effects with genuine epistasis present: a good strategy should produce results better than the linear model, as close as possible to the nonlinear model without this strategy. This would indicate that the model was able to pick up the genuine epistatic nonlinearity to produce a better performing PGS than a linear model, while preserving the signal.

(2) Simulated phenotype involves joint tagging effects with no genuine epistasis present: a good strategy should produce results for the nonlinear model as similar as possible (but no worse) as the linear model, as there was no real nonlinear signal.

Table 1 demonstrates how the "**SNP-dosage weighting**" strategy satisfies these criteria. Row 2 shows that when there is genuine epistasis without joint tagging effects, then when applying this mitigation strategy the nonlinear NN model is still able to capture the bulk of the performance gain over the linear NN model: 9.5% vs 10.9% without the mitigation strategy. Whereas row 5 demonstrates that when there is no epistasis but joint-tagging effects are present, the performance difference between the linear and the nonlinear models with this strategy is not significant ($p = 0.398$).

In addition to these clarifications, for this revision we also added the new mixed, additive+epistatic, simulation scenarios with this "**SNP-dosage weighting**" strategy enabled, whose results are shown in Table 1, rows 7-8. As the results for scenarios were as expected, intermediate between the 'pure' 100% epistatic and the 'pure' 100% additive scenarios, this adds further evidence that our mitigation strategy is viable and appropriate.

Taken together, given these results, we believe that the efficacy of the "**SNP-dosage weighting**" mitigation strategy is clear. Ultimately, of course, each reader will have their own use case which is not possible to know beforehand thus the practicality and incremental benefit of SNP dosage weighting will vary. We have also now added this to the limitation part of the Discussion:

"Our paper introduced two different mitigation strategies aimed at controlling the LD-driven 'joint tagging effects'. The practical benefits of the "SNP-dosage weighting" strategy will vary based on each individual dataset. On the other hand, the "LD-clumping + distance filtering" strategy is not practical in generating PGS, as

it is a brute-force approach that eliminates most of the signal in its distance filtering step, therefore, it is only useful to show as a point of reference."

"Reviewer #1 (Remarks on code availability):

Choice of Python Library Restricts Code Accessibility: Although the provided code appears functional, the authors use the niche Python library "knet". This choice limits accessibility and reproducibility for readers unfamiliar with this framework. Using more widely adopted libraries like TensorFlow or PyTorch would improve usability and facilitate peer review."

We agree with the reviewer regarding usability/accessibility/reproducibility. We clarify that our model is indeed implemented entirely in Pytorch (please see `knet_main_pytorch.py` / `knet_manager_pytorch.py`). In this case, 'knet' is just a wrapper for the code used by our models.

"Reviewer #2 (Remarks to the Author):

This is an interesting paper, investigating the use of NN based methods in the generation of polygenic scores, and polygenic risk models that include other risk factors, to predict diseases or phenotypes in UK biobank.

I think this paper is well written. It is has a more methodological focus than some applied papers, however I found it very easy to read, and that the authors had done a very good job of making potentially complex genetic concepts understandable. I think the paper is important as it highlights that novel NN methods that have offered improvements in many areas of science, have some potential trade offs for genetic analyses and may not be currently suitable for PGS generation at scale, and the authors do a good and balanced job of describing this."

We thank the reviewer for appreciating the amount of effort we put into our paper.

"The authors rigorously generate PRS, and NN models, to predict 28 phenotypes in UK biobank. They compare NN models that potentially account for nonlinear interactions (including gene-gene interactions and gene environment interactions). They first perform an analysis in a simulated dataset that compares traits that are best explained entered by a simple additive model of heritability, traits that include joint tagging interactions, and traits whose variance is explained by true epistasis (phenotypes that are at the opposite end of the spectrum - explained entirely by complex non linear interactions). I enjoyed this approved of simulating a positive and negative control dataset to check that their approach could distinguish between these two artificially created extremes, before then preceding to an analysis of real data from 28 UKBB phenotypes.

Their analysis when comparing within the two NN approaches highlighted that the joint tagging effect is a very important consideration when considering whether observed statistical epistasis relates to joint tagging effects or genuine epistatic interactions.

I found the noise introduced by the NN approach compared to baseline models interesting. They highlighted that the baseline linear PGS models were more predictive than the NN approaches they used. This is an important “negative” and is reported clearly, I think this is an important study and message for the field.”

We very much value the reviewer’s observation that negative results can be important to keep overinflated expectations in check, especially nowadays with respect to AI.

“I think the work supports the claims the authors make, the methods are sound, with some potential limitations,

Comments/questions

General

The authors highlight that they had certain selections criteria for traits to be included, is it possible that the traits and the limited population selected do not cover the full breadth of variation in disease heritability - and some diseases may have more GG or GE interactions than others, where NN models may have more relevance in helping elucidate genetic architecture.”

We completely agree with the reviewer on this point. Given the vast scope of variation in disease heritability (inclusive of GxE), we believe it is unavoidable that this will need to be a limitation of our more focused deep learning study. We now better acknowledge this point as a principal limitation in the Discussion:

“Of the 28 examined traits that were available in the PGS Catalog and met our QC criteria, 21 were cancer phenotypes, whose genetic architecture may differ from other phenotypes and diseases that arise from normal trait variation due to the importance of somatic mutations for cancers⁴¹”

The traits we could use for our study were limited by the data availability in the PGS Catalog that also met our strict QC criteria. This resulted in a bias towards selecting cancer phenotypes, some of which may have different genetic architectures relative to other traits. However, out of the 28 evaluated traits seven were non-cancer of diverse aetiologies, including five cardio-metabolic traits (ischemic stroke, venous thromboembolism, type 2 diabetes, fasting glucose and HbA1c) and two others: height and major depressive disorder. As such, our conclusions are still expected to be broadly applicable to the (medical) phenome.

“Can they make general inferences about using linear PGS models v NN models based on these results? Are there criteria that would need to be met for NN models to outperform simpler methods, or do their results from simulated data highlight that even in the most extreme settings of genetic non linear interactions, there is no prediction advantage to their NN approach?”

In our simulations of generic quantitative traits, the nonlinear NNs do actually outperform linear ones, although only by ~10%. This is an important result, as it shows that if there is real nonlinearity, i.e. genuine epistasis, then with a sufficiently large sample size, there is a potential advantage for nonlinear NNs. Whether this ~10% advantage is substantial, is relative however. Given the costs and the sample sizes it would require it may not be worth it in all cases. However, adopting a more optimistic perspective, we recall that PRS-CS only outperformed the original LDpred1 algorithm by less than 10% ([nature.com/articles/s41467-019-09718-5](https://www.nature.com/articles/s41467-019-09718-5)), so what one considers a substantial improvement is relative. We of course have to remind the reader that this ~10% advantage was only observed in simulations, in real data experiments NN models proved inferior to additive PGS baselines. So our view is that the real likely limitation of NN-based PGS is that there may not be (enough) nonlinearity in the population-level trait variance to make them worthwhile.

“How generalisable are their findings?, and how fragile are the findings to their choice of parameters, discovery data, covariates, phenotypes, genetic data, ancestry etc....I think they could comment on this a little more in the limitations, or show more data to back up the generalisability. this is important in particular because the strength of their negative findings depends a little on the generalisability of their findings.”

The reviewer brought up an important question about generalisability. While we used all the data that met our QC criteria we could find, more data, particularly data from ancestrally more diverse populations would have been interesting, a limitation that we have now added in the Discussion section at:

“Another limitation of our study was that all individuals in our analyses were of European ancestry, which was due to the poor representation of more diverse populations in the UKB. Finally, in the near future, the quantity of genomic data is expected to grow and reach the scale of tens of millions of individuals, including better representation of more diverse populations, also capturing their whole genome sequence at ever greater read depths^{42,43}.”

The reviewer also enquired about our model/parameter selection choices. A full exploration of model architectures would have involved a random search in a very large parameter space, typically requiring ~50 trials per scenario, which would have been infeasible for the simulations and 28 real traits in our study. However, during the first author's PhD work, a larger range of parameters were evaluated (including adding

convolution layers), and there it was found that the best performing models were always the simplest, shallow MLP architectures with very few layers. This is consistent with the expected result when the amount of non-linearity is low/non-existent, as that is when the NN model gets closest to a basic linear regression model (one layer, one neuron). The details of this can be found in Chapter 4 of Dr Kelemen's thesis (<https://doi.org/10.17863/CAM.72055>), which was also corroborated by others (<https://pmc.ncbi.nlm.nih.gov/articles/PMC6218236/> and <https://pubmed.ncbi.nlm.nih.gov/35072137/>), who found that simpler architectures worked better (usually meaning less poor). To provide justification for our model choices in this paper and to help the reader assess the likely generalisability of our findings to their particular use case(s), we wrote a detailed explanation that cites and reviews the relevant literature for our model choices in the supplementary ("**Supplementary Note 2: parameter & model selection for NNs for PGS generation**") which is also referenced in the main text in the Methods section under "**Neural-network models**".

"They used standard criteria to QC data and exclude covariates before model generation, however is there a danger that this could remove the chance for the NN models to find occult interactions in the data, not apparent during the QC procedures? I think this is computationally efficient, but could it also reduce the change of finding new non-intuitive interactions within the data?"

The reviewer brings up a valuable perspective. It is true that every excluded predictor is a potential 'hit' for association. However, this is a necessary tradeoff in order to remove the target phenotype's signal from the covariates used to predict it. The alternative would have been misleading, as one can imagine the effect of keeping predictors that overlap partially or wholly with the target outcome. (For example, modelling BMI while having weight in the model). In any case, typically, only 2-3% predictors were excluded this way (out of the 198), which we believe is a small tradeoff.

"Did you consider other non NN based approaches to take non linear effects into account, given the good performance of baseline PGS models, could you consider approaches that build on the baseline methods?"

We thank the reviewer for this suggestion. However, the focus of our paper was to evaluate the use-case for NN-based approaches for PGS generation. Specifically, we wanted to examine if the nonlinear capacity of NNs could be harnessed to improve over additive baselines. The field of machine learning is moving very fast with new techniques developed at a rapid pace. Therefore, a comprehensive 'bake-off' style project to evaluate all current methods, extending or combining different baselines, would have been infeasible and beyond the scope of our ambitions here. We believe that our readers will appreciate the tighter focus of our paper as it currently stands but we now also acknowledge this as a limitation in the Discussion by adding the following text:

“Alternative avenues by which NNs may contribute to phenotype prediction could be through combining multiple models in ensemble methods or through generating annotations (such as the presence of sequence features) that could be used for PGS generation.”

“Page 5

Comparison of linear and nonlinear NN models when prediction simulated polygenic scores with entirely additive effects. Can they just explain again what the main result is here: “This result in turn suggests that non linear NN models are better able to infer (false) interaction effects that capture phenotypic variance than they are at inferring (true) additive effects.” I think non linear NN models seemed to offer more improvement over linear models even when there were no introduced introduced epistasis or joint tagging effects. I am not sure what the meaning of this is, and wonder if the authors could rephrase or re explain?”

We are very grateful to the reviewer for bringing the lack of clarity of this section to our attention. We rewritten this section to now state:

“The fact that nonlinear NNs outperform linear NNs, even with no genuine epistasis or no joint-tagging effects present, has important implications for interpreting the results of the other scenarios. In the two scenarios where the phenotype signal arose entirely from genuine epistasis, with and without our mitigation strategy applied (10.9% and 9.5%), only the latter, smaller value should be considered to be measuring genuine epistasis. The difference between these two results in turn should then be interpreted as the nonlinear NNs ability to exploit LD to better model additive effects in the absence of any epistasis.”

“Figure 3

They report the average change in polygenic score performance when using NN models compared to PGS baseline models, there is a spread, with some NN models (particularly two outliers in the models including covariates) seeming to perform considerably better than the PGS baseline, is this just noise or does it reflect the fact that in a select subset of diseases,”

The reviewer’s comment appeared potentially truncated in the version we received; however, we interpret it as asking if there was a subset or traits where the nonlinear NN models performed substantially better than the baselines. In the right side of Figure 3 (the SNP+Cov models), it may appear at first that two phenotypes are outliers. However, upon closer inspection, we believe that this is not the case as the relevant comparison is

not only against the baseline regression method (i.e. '1' on the Y-axis), but also between the paired results of the nonlinear vs linear models (blue vs orange).

Taking this perspective, it can be seen that while yes, these two results are substantially better than the regression model (ie far from '1' on the Y-axis), however, the difference between the nonlinear and linear models was modest, as the difference between the matching orange and blue dots was quite small. To clarify this, we have also modified the figure to connect the paired results with line segments. We reproduce this figure below with additional highlighting to illustrate this:

"#Reviewer #4 (Remarks to the Author):

This paper presents a comparison of linear and non-linear neural network models for genomic prediction in humans. No large advantage if any is found on using non linearities, overall. I cannot see a clear target of the study. If the goal is to assess NN performance, at least a linear standard method should be added, such as ridge regression. If the goal is to compare linear vs say epistasis, linear model such as GBLUP can be generalized to include a Ax_A matrices."

We thank the reviewer for pointing out the lack of clarity on our part with regards to the baseline methods testing. We can clarify that all our NN models started off from summary statistics versions of the baselines the reviewer suggested, for example

LDpred or PRS-CS, which equal or outperform the standard linear models mentioned, like ridge regression GBLUP. This is a crucial first step of our “SNP-dosage weighting” strategy, as we argue that by using such weights we can account for any ‘false’ epistasis due to joint-tagging effects. However, what was not emphasised adequately in our initial text was that as all NN results were expressed as a % of these baseline PGS methods, this comparison was already implicitly present in our results. If an NN result was less than 100%, then that means that the NN model performed worse than the baseline PGS, whereas if it was greater than 100%, then that meant that it beat the established baseline. This was briefly mentioned under **“Quality control of real data results”**, and the full list of baseline methods is mentioned under the Methods section, under the heading **“Selection of phenotypes and generating the PGS”**:

“...were constructed using an LD-aware PGS method. We defined LD-aware as a method that incorporated LD reference panels to model signal overlaps between markers due to LD. The full list of these methods was: DBSLMM⁴⁶, metaGRS⁴⁷, lassosum⁴⁸, LDpred³⁰, LDpred2⁴⁹ and PRS-CS⁵⁰.”

However, we recognise that the fact these comparisons were performed was not necessarily clear to the reader. Therefore, we have now explicitly spelled out these in our results section by adding the following text:

“All results are expressed relative to the LD-adjusted PGS, which means that negative values (for simulations) or values below 1.0 (for real data) indicate an unfavourable comparison between the NN models and their respective PGS baselines.”

The legend for Fig 3 now states:

“...expressed as a fraction of their baseline linear regression models. Therefore, a value below 1.0 indicates that the NN model was outperformed by the LD-aware additive baseline.”

And finally, the legend of Table 1 now also states:

“Positive values indicate that the nonlinear model performed better than the linear model and conversely, a negative value indicates that the baseline additive PGS performed better.”

“Methods are not clearly described and, sometimes, conclusions are reached without enough evidence. I agree that linkage disequilibrium can be confounded with epistasis, but a comparison between linear and non-linear NN does not provide a clear conclusion between what is more prevalent. In fact, both can be present simultaneously.”

The SNP dosage weight is not justified. As I understood initially, it required running an algorithm twice, the first time to obtain the weights (I assume only with the training set). It seems these are weights for SNPs obtained from previous studies, but how are these weights obtained, are of on same dataset? same diseases. If so, you will likely be using test and training data even if unknowingly."

We believe that the framework described in our Methods section, under the headings **"Selection of phenotypes and generating the PGS"**, **"Controlling for joint tagging effects"**, **"Evaluating the evidence for nonlinearity"** and **"Simulation scenarios"** provides sufficient details to address these concerns.

Under **"Selection of phenotypes and generating the PGS"**, we state:

"...phenotypes were selected by scraping the PGS Catalog⁴⁵ database with the following criteria. Eligible PGS needed to have a training set of at least 50,000 individuals that excluded the UKB..."

Therefore, we are confident that the potential concern about data leakage/overfitting has been minimised.

Next, under **"Controlling for joint tagging effects"**, we described in detail how the application of our SNP-dosage weighting approach accounts for any extra performance gains due to incomplete coverage and LD when comparing the linear and nonlinear NN models:

"...when comparing the performance of nonlinear and linear NNs, where both models had these per-SNP PGS weights available, any advantage of the nonlinear NN over the linear NN would be likely due to genuine epistasis..."

Additionally, under **"Evaluating the evidence for nonlinearity"** we provide detailed derivations for inferring nonlinearity. Here, we show how by removing the nonlinear capacity of a NN, but otherwise keeping its architecture identical, comparing performance between the nonlinear and linear NN models one can infer the existence of genuine epistatic effects.

Finally, under **"Simulation scenarios"** we provide evidence that by combining the aforementioned techniques we are able to distinguish between joint tagging effects and genuine epistasis. We would point the reviewer to Table 1, where rows 1, 2, 5 and 6 provide strong evidence of the validity of our strategy. Comparing rows 1 and 2 demonstrates that when there are genuine epistatic effects, the application of the SNP-dosage weighting approach preserves the bulk of this nonlinear signal (10.9% vs 9.5%). Comparing rows 5 and 6 demonstrates that when there was no genuine epistasis but there were confounding joint tagging effects, then naively testing for nonlinearity will

result in the detection of significant ($p=2.725 \times 10^{-4}$) epistatic effects. On the other hand, by applying our mitigation strategy, the difference between the nonlinear and the linear NN models became non-significant ($p=0.398$), suggesting that when there is no real epistasis, our model won't think that there is, even if there were confounding by joint tagging effects present.

Taken together, these results suggest that if there is genuine epistasis, our model will be able to find it, and if there is no genuine epistasis it won't think that there is, regardless of the confounding from joint-tagging effects.

"But even so, what are the properties of this approach? May be an ensemble approach would be more clearly justified..."

The focus of our paper was to evaluate the use-case for NN-based approaches for PGS generation. Specifically, we wanted to examine if the nonlinear capacity of NNs could be harnessed to improve over additive baselines. The field of deep learning is moving very fast with new techniques developed at a rapid pace. Therefore, a comprehensive 'bake-off' style project to evaluate all current methods, including their ensemble versions, would have been infeasible and beyond the scope of our ambitions here. We believe that our readers will appreciate the tighter focus of our paper as it currently stands but we now also acknowledge this as a limitation in the Discussion by adding the following text:

"Alternative avenues by which NNs may contribute to phenotype prediction could be through combining multiple models in ensemble methods or through generating annotations (such as the presence of sequence features) that could be used for PGS generation."

"Additional comments

- Reference 45 to PGS catalog cannot be accessed."

It may have been a temporary connectivity issue. We can confirm that the link to the PGS Catalog is operational.

"- It seems multi layer perceptrons are the only strategies used, what about CNNs?"

CNNs perform well when the underlying patterns fit the model's assumption: regular spatial structures, such as features in pixel data. However, SNP data does not follow such regular patterns, as the distance between the loci SNPs refer to is irregular, and SNPs only represent deltas from a reference, rather than independently meaningful patterns. During the first author's PhD work, a larger range of parameters were evaluated (including adding convolution layers), and there it was found that the best

performing models were always the simplest, shallow MLP architectures with very few layers. This is consistent with the expected result when the amount of non-linearity is low/non-existent, as that is when the NN model gets closest to a basic linear regression model (one layer, one neuron). The details of this can be found in Chapter 4 of Dr Kelemen's thesis (<https://doi.org/10.17863/CAM.72055>), which was also corroborated by others (<https://pmc.ncbi.nlm.nih.gov/articles/PMC6218236/> and <https://pubmed.ncbi.nlm.nih.gov/35072137/>), who found that simpler MLP architectures worked better (usually meaning less poor).

An additional problem with using CNNs (or other more complex layer types), is the difficulty in maintaining the apples-with-apples type of comparisons for our nonlinearity inferring algorithm. As described, we infer nonlinearity by comparing the performance of the nonlinear vs linear NN models, where the linear models differ from the nonlinear models only by switching the nonlinear activation functions off, but keeping their architectures otherwise identical. However, in addition to these nonlinear activation functions, CNNs derive their nonlinear capacity from maxpooling layers. Therefore, it would not have been possible to use CNNs in our framework, as for the linear NNs maxpooling layers would have had to be either removed or substituted by other layer types (such as average pooling). Therefore, if we had used CNNs, we would not have been able to claim that our framework would be able to quantify the performance gains due to nonlinearity.

As another reviewer also raised the issue of parameter/model selection, we added a detailed explanation that cites and reviews the relevant literature into the Supplementary ("**Supplementary Note 2: parameter & model selection for NNs for PGS generation**"). This section now provides justification why simpler, MLP-type NNs were chosen in this work instead of other more complex models.

"- Methods are not enough detailed , a github or similar with documented code would help. Reviewer #4 (Remarks on code availability): No code available."

The codebase of our project was available under the master branch of the github link (github.com/mkelcb/dl-prs-paper). This was, however, not set to be the default branch, so the reviewer may not have seen this. We thank the reviewer for drawing our attention to this. We rectified this issue now.